# Research

behaviour, ecology

cooperation, social behaviour,
helping behaviour, socio-economic status

**Author for correspondence:**
Nichola Raihani
e-mail: nicholaraihani@gmail.com

# Neighbourhood wealth, not urbanicity, predicts prosociality towards strangers

Elena Zwirner[1] and Nichola Raihani[2]

[1]Genetics, Evolution and Environment, University College London, Gower Street, London WC1E 6BT, UK
[2]Department of Experimental Psychology, University College London, 26 Bedford Way, London WC1H 0AP, UK

NR, 0000-0003-2339-9889

Urbanization is perhaps the most significant and rapid cause of demographic change in human societies, with more than half the world's population now living in cities. Urban lifestyles have been associated with increased risk for mental disorders, greater stress responses, and lower trust. However, it is not known whether a general tendency towards prosocial behaviour varies across the urban–rural gradient, or whether other factors such as neighbourhood wealth might be more predictive of variation in prosocial behaviour. Here, we present findings from three real-world experiments conducted in 37 different neighbourhoods, in 12 cities and 12 towns and villages across the UK. We measured whether people: (i) posted a lost letter; (ii) returned a dropped item; and (iii) stopped to let someone cross the road in each neighbourhood. We expected to find that people were less willing to help a stranger in more urban locations, with increased diffusion of responsibility and perceived anonymity in cities being measured as variables that might drive this effect. Our data did not support this hypothesis. There was no effect of either urbanicity or population density on people's willingness to help a stranger. Instead, the neighbourhood level of deprivation explained most of the variance in helping behaviour with help being offered less frequently in more deprived neighbourhoods. These findings highlight the importance of socio-economic factors, rather than urbanicity *per se*, in shaping variation in prosocial behaviour in humans.

## 1. Introduction

The world's population is expected to rise from 7.7 billion to 9.7 billion inhabitants between now and 2050 [1] and most of this global population increase will be absorbed by cities. City life is associated with higher incidence of physical ailments (e.g. allergies, heart disease, diabetes, and cancer [2–4]) and mental health problems (e.g. stress, anxiety, and schizophrenia [5,6]). In addition, city-dwellers are often perceived as being less cooperative than their rural-dwelling counterparts [7–17]. For instance, studies have variously found that people living in urban locations are less likely than those living in nonurban locations to complete and return a postal survey [10], to help a stranger in distress [18], to correct an accidental overpayment in shops [19], or to donate to charity [20]. Nevertheless, the picture appears to be quite mixed. In one meta-analysis, urban residents were more likely than nonurban residents to help strangers in 9/46 studies and no difference was found in 10/46 studies [21]. In another study, people living in urban locations were more likely to both offer and receive help from friends compared to those living in rural or nonurban locations [7], and other work has also reported increased return rates of lost letters from cities compared to rural communities [22,23]. Here, we present results from three real-world experiments conducted across 37 neighbourhoods in 12 UK cities and 12 towns and villages, asking whether tendency to help a stranger varies with (i) urbanicity, or (ii) neighbourhood indices of deprivation; and (iii) what other situational variables affect whether help will be provided.

Living in cities could reduce the willingness to help strangers for several reasons. For example, people living in cities may experience a faster pace-of-life and increased perceptual load [14], both of which could make people less likely to offer help, either because they do not note that help is needed or do not have time to offer help. The greater population size in cities might also elicit a bystander effect, whereby individuals are less likely to offer help because they do not feel personally responsible for providing assistance [24], but see [25]. Indeed, previous work has shown that people are more likely to feel personally responsible and to help in situations where they are directly asked, compared to situations where no request is made [26–29]. We tested this effect in two of our three experiments, and additionally asked whether rates of helping under direct request (compared to no request) varied across urban and nonurban locations. City life might also reduce the willingness to help strangers because people have a higher number of one-shot encounters with strangers, where any downstream consequences of prosocial action (or inaction) are less likely to be realized. Indeed, individuals are apparently less likely to behave prosocially in one-shot settings when they cannot be identified by others [30–33]. In two of the experiments below, we tested whether people were more likely to help a stranger when they were accompanied by a friend or acquaintance, compared to when they were alone; and whether the audience effect had differential effects on help in cities versus towns.

There are also important confounding factors associated with urbanicity that could lead to variation in prosocial behaviour. For example, indices of deprivation and socio-economic status vary widely within cities [34–37] and it could be variation in these parameters, rather than urbanicity *per se*, that predicts variation in social behaviour [38]. Work done in two similar-sized neighbourhoods of a single UK city found that crime and antisocial behaviour (littering) were more common in the more deprived of the two neighbourhoods [36], and that people from the more deprived neighbourhood also donated less money to a partner in an experimental Dictator Game, and were more willing to cheat by stealing money from a partner in a different experimental task [37]. Another study in a single US neighbourhood found similar results, with people from wealthier neighbourhoods being more likely to self-report prosocial behaviour and to return a lost letter [39]. Lost letter experiments performed in different neighbourhoods across London [35] and in an urban and rural location in Australia [34] have found similar effects, with letters being returned more often from higher-wealth neighbourhoods. In a cross-cultural study of helping behaviour in 23 cities around the world, Levine *et al*. [40] showed that helping behaviour was inversely associated with the country's economic productivity, whereas in another study of 24 US cities, helping behaviour was positively related to average purchasing power [41].

Notwithstanding the studies cited above, much previous work exploring the effects of urbanicity on prosociality has tended to run experiments in the centre of the city or town (e.g. [8,17,23,40,41]) and have also used a single indicator of deprivation or wealth for an entire city, thereby glossing over important within-city differences in deprivation that could be associated with residents' willingness to help a stranger in need. Given these confounding factors, it is not yet clear whether urbanicity does indeed affect prosocial behaviour or whether previous results have confounded urbanicity with

socio-economic variables that vary across the urban–rural continuum [42]. We address these issues here.

We conducted three experiments in 37 neighbourhoods (across 12 cities and 12 towns and villages in the UK). Our experiments measured whether people would; (i) post a lost letter back to the experimenter; (ii) help the experimenter pick up some dropped items; and (iii) allow the experimenter to cross the road. By using three different help measures, we aimed to reduce the possibility that any results are idiosyncratic to one particular type of help. We expected that people would help more in towns than in cities, and would help more in higher-wealth neighbourhoods than in lower-wealth neighbourhoods. For the lost letter experiment and the dropped item experiment, we ran two conditions where targets were either directly asked for help or were not directly asked (see Material and methods). We expected help to be more common when targets were directly asked for help [26–29]. For the dropped item experiment and the road-crossing experiment, we recorded whether targets were alone or with another adult when the help scenario was triggered. We expected that people would be more helpful in the presence of a salient audience (i.e. a friend/acquaintance) compared to when they were alone.

## 2. Material and methods

### (a) Experimental locations

All field experiments were conducted in cities and towns/villages (hereafter 'towns') of mainland United Kingdom (hereafter UK), over July–September 2014, May–October 2015, and May–July 2016. We used the 2011 Census Data (available at: www.ons.gov.uk) to select 12 cities and 12 towns based on their population size and density. We selected cities on the basis of having more than 100 000 residents (following [21]). Specifically, we created a pre-sample list of the 24 most densely populated cities in the UK, and selected the top six and bottom six from this list. We deliberately excluded London as it is the only 'global' city in the UK [43]. The average population size of the cities included in this dataset was 446 600 (±60 348), with cities ranging in size from 220 570 to 995 480 inhabitants. We selected towns on the basis of having fewer than 20 000 residents (the point at which previous work has reported a decline in helping behaviour [8]). We chose towns that were easily accessible from the cities that we sampled, either by train or bus. The mean population size of the towns in this study was 12 126 (±1581), with towns ranging in size from 2998 to 19 656 inhabitants. Cities had significantly larger population sizes than towns ($t$-test, $t = 7.38$, d.f. = 11, $p < 0.001$).

In addition to measuring population size, we also consider population density because some of the proposed mechanisms by which urbanicity is thought to reduce prosociality (e.g. perceptual load and diffusion of responsibility) are more likely to be affected by the density of a population rather than its size [14,23,44,45]. We measured population density at the neighbourhood level, using data from the Office for National Statistics (England and Wales) and the 2011 Census (Scotland). Data for England and Wales came from the 2015 census, whereas data for 2011 were available for Scotland. There was no difference in the area of the neighbourhoods (km$^2$) sampled in the cities and towns ($t$-test: $t = 0.99$, d.f. = 15, $p = 0.34$) but there was a significant difference in population density (number of people per square kilometre) of towns and cities in the neighbourhoods

we sampled (mean density city = 3961 ± 463, mean density town = 2136 ± 420; $t$-test: $t = 2.92$, d.f. = 22, $p = 0.008$).

## (b) Neighbourhood wealth

In each city, we ran experiments in a higher-wealth and a lower-wealth neighbourhood, to allow us to disentangle the potentially separate effects of neighbourhood wealth from urbanicity on prosocial behaviour. We selected neighbourhoods using the 2011 UK Census (these are the smallest areas for which data are available and are referred to as lower super output areas—LSOAs—in England and Wales and Data Zones in Scotland) based on their index of multiple deprivation (IMD) score. The IMD is an index consisting of multiple factors affecting deprivation of an area, each of which is given a percentage weighted value. The principal factors of the index are income and employment [46], which together count for approximately 50% of the total weight on the IMD score. Within each city, we selected one higher-wealth and one lower-wealth neighbourhood, according to the upper and lower quartile of the IMD distribution, respectively. For all-but-one of the towns we sampled, there was insufficient variation in within-town indices of deprivation to sample a higher-wealth and a lower-wealth neighbourhood within the same town. In one town (Helensburgh), there was sufficient neighbourhood-level variation in wealth and in Helensburgh we, therefore, ran experiments in a higher-wealth and a lower-wealth neighbourhood. For the remaining 11 towns, we ran experiments in one neighbourhood per town (six higher-wealth, five lower-wealth neighbourhoods). The total number of neighbourhoods sampled in this paper is, therefore, 37, which derive from 12 different towns and 12 different cities across the UK. Twenty-four of these neighbourhoods were in cities and 13 in towns. Of these neighbourhoods, 18 were categorized as 'high-wealth' and 19 as 'low-wealth'.

For the 18 locations we sampled in England, we compared the variance in indices of multiple deprivation for each LSOA in that area using publicly available data from UK Government websites. These analyses are restricted to the English LSOAs because the way that IMD is calculated varies slightly across the four countries of the UK. As expected, variance in IMD was greater for the cities we sampled than for the towns (variance cities = 359.4, variance towns = 256.5, Levene's test for homogeneity of variance, $p < 0.001$). LSOAs within cities were also significantly more deprived than LSOAs within towns in this sample (mean IMD city = 33.2, town = 25.4, 2 sample $t$-test, $t = 6.53$, d.f. = 231.3, $p < 0.001$).

## (c) Helping behaviour

We used three help measures to examine cooperative tendency across urban scales: posting a lost letter [47], helping to pick up dropped items, and allowing a pedestrian to cross the road. For the dropped item and road-crossing measures only one investigator (EZ) conducted the trials (with help from two observers). In one neighbourhood only, the lost letter experiment was conducted by a second investigator. As this experiment does not involve a face-to-face interaction with the experimenter, we do not believe this to be problematic.

### (i) Lost letters

A total of 879 stamped letters addressed by hand to 'E. Zwirner' at a PO box address were dropped in 37 neighbourhoods between July–September 2014 and May–October 2015. Following Holland *et al.* [35], we used only the initial of the addressee's name such that subjects would not know whether the letter was being posted to a male or a female recipient. To test the effect of a direct versus indirect help request, 439 of the letters were dropped on the pavement with the address facing up on rain- and wind-free mornings, whereas 440 letters were left on car windscreens (following [47]) with a post-it saying: 'Could you post this for me please? Thank you'. For the indirect requests, letter drop points in the neighbourhood were randomly determined using Google Maps (www.google.com/maps) and were never on the same street as a postbox or where a postbox was visible.

### (ii) Dropped items

The dropped item experiment was conducted 398 times in 37 neighbourhoods, between July–September 2014 and May–October 2015. The procedure began with EZ walking with a handful of 20 cards on the pavement. A pedestrian passing on the same side of the street was selected to be a subject if they appeared to be 18 years or older, was not carrying items such as bags or a phone in their hand, and had no physical handicap. When the subject was approximately 5 m away, EZ dropped the cards onto the pavement, bent down and began picking them up one at a time. We conducted two experimental treatments to measure variation in the tendency to help pick up the dropped items: (i) direct request ($n = 174$ observations): after dropping the cards, EZ bent down to retrieve the cards and also looked at the subject and asked: 'Could you help me, please?' (ii) Indirect request ($n = 224$ observations): after dropping the cards, EZ bent down to retrieve the cards and looked at the subject, but did not ask for help. After picking up the envelopes, we recorded whether the subject stopped to help with the dropped items, and whether he/she was alone or with an acquaintance.

### (iii) Road-crossing

The road-crossing experiment was conducted 90 times in 26 neighbourhoods between July and September 2014. The procedure began with EZ standing on the pavement. An approaching car was selected if its speed was estimated to be below the speed limit and if no other car was present behind it. When the selected car was approximately 10 m away, EZ started to cross the road. If the car slowed down/stopped the investigator continued to cross the road, if it did not, EZ stepped back on the pavement. After the attempt to cross the road, we recorded whether the car allowed the pedestrian to cross, and whether the driver was alone or with a passenger.

## (d) Statistical analysis

Data were analysed using R v. 3.6.2. Data and code to reproduce analyses are available online at https://osf.io/cmdfk/. To explore the main question of interest (whether urbanicity, population density, or neighbourhood wealth affected the probability of receiving help), we used a Bayesian approach with the statistical packages rethinking [48] and stan [49]. We fitted a single model to the full dataset ($N = 1367$ observations) containing all terms of interest (Urbanicity, Population Density, Neighbourhood Wealth) as a logistic regression with 'Help' specified as the binary outcome variable. Neighbourhood Wealth and Urbanicity are binary terms (1 = high wealth and 1 = urban location, respectively).

*Proc. R. Soc. B* **287**: 20201359

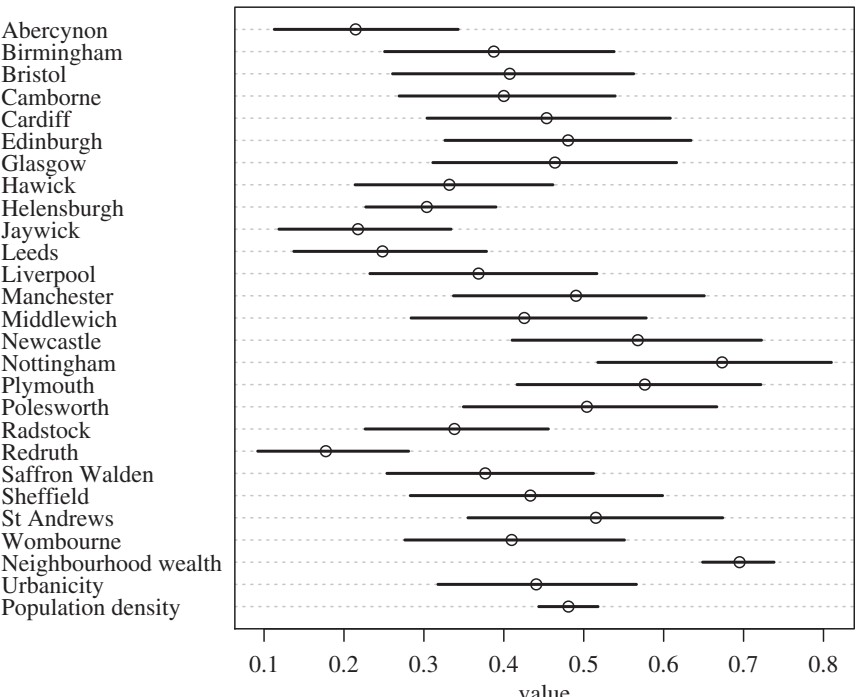

**Figure 1.** Precis plot showing posterior mean binomial probabilities (with 89% percentile intervals) for (i) each location's intercept and (ii) the slopes associated with Urbanicity, Population Density, and Neighbourhood Wealth. Percentile intervals show the interval within which 89% of the probability mass for the predicted means is found. Following [48], we use 89% intervals to avoid readers drawing a spurious inference that these intervals correspond to significance tests. For each location, the intercept denotes the estimated probability of receiving help in that place (where 0 = low probability and 1 = high probability). For the parameters (Urbanicity, Population Density, and Neighbourhood Wealth), the binomial probability indicates the estimated effect of each treatment on the binomial probability of receiving help. Thus, the plot shows that in high-wealth neighbourhoods, there is a high probability of receiving help, but that there is little appreciable effect of urbanicity or population density on the probability of receiving help (binomial probability close to 0.5 for both parameters indicating that these parameters do not affect the likelihood of receiving help above chance levels).

Population Density is a continuous variable, which we log-transformed (and scaled) since we did not expect help to scale linearly with population density. In addition to the explanatory terms, we included a random intercept for place (24 levels, one for each city/town). Following recommendations for logistic regression models [48], we used weak priors of (0, 1.5) and (0, 0.5) (mean, standard deviation) for the random intercepts and explanatory terms in all models, respectively. These priors are concentrated on low absolute differences and are more appropriate than standard flat priors when the differences between levels of a treatment are expected to be small. Maximum posterior estimates for coefficients included in the model were determined using map2stan and Markov Chain Monte Carlo estimation, as described in McElreath [48]. We initially tested the model for robustness using 4 chains, 250 iterations, and 50 warmups. Estimates were then derived from a model using 1 chain, 2500 iterations, and 250 warmups. Analyses of the effects of direct requests and audience presence on helping behaviour were performed using $\chi^2$-tests.

## 3. Results

In total, we recorded 1367 instances where a member of the public decided whether or not to help the experimenter. Help was offered on 643 (47.0%) of all occasions. Breaking down these numbers by help type, 485/879 (55.1%) letters were returned, 130/398 (32.7%) people helped the experimenter to pick up some dropped items, and 28/90 (31.1%) cars stopped

to let a pedestrian cross the road. The main variable influencing whether help was offered across all experimental conditions was neighbourhood wealth (probability of receiving help in high-wealth compared to low-wealth neighbourhood: 0.70, 89% intervals: 0.64, 0.74, figures 1 and 2). There was no evidence to suggest that urbanicity or population density affected the chance of receiving help (figure 1). We examine the other situational factors affecting propensity to help in each condition separately below.

### (a) Direct/indirect request

We found mixed support in favour of our prediction that direct requests would be more successful at eliciting help. In the dropped item experiment, 82/174 (47.1%) pedestrians who were directly asked helped the experimenter to pick up dropped items, compared to 48/224 (21.4%) who were not asked ($\chi^2$-test, $\chi^2 = 28.2$, d.f. = 1, $p < 0.001$). Nevertheless, letters posted under windscreen wipers with a note (direct request) were not more likely to be returned than letters left on the street (245/440, 55.7%, letters posted on windscreens were returned, compared to 240/439, 54.7%, letters left on the street; $\chi^2$-test, $\chi^2 = 0.05$, d.f. = 1, $p = 0.82$). Overall, there was no effect of urbanicity on the likelihood of receiving help either in the direct request condition (City: 223/403, 55.3%, requests helped; Town: 104/211, 49.3%, requests helped; $\chi^2$-test, $\chi^2 = 1.80$, d.f. = 1, $p = 0.18$), or in the no request condition (City: 199/436, 45.6%, non-requests helped; Town: 89/227, 39.2%, non-requests helped; $\chi^2$-test, $\chi^2 = 2.262$, d.f. = 1, $p = 0.13$).

**Figure 2.** Percentage of occasions where help was received by (*a*) posting a lost letter, (*b*) retrieving dropped items, and (*c*) allowing a pedestrian to cross the road. Dark bars show high-wealth neighbourhoods; light bars show low-wealth neighbourhoods. Agresti–Coull confidence intervals are displayed.

## (b) Audience effects

Counter to our expectations, anonymity did not reduce people's willingness to help a stranger. There was no difference in tendency to stop to let a pedestrian cross the road based on whether the driver was accompanied by a passenger (16/46, 34.8%, stopped) or not (12/44, 27.3%, stopped, $\chi^2$-test, $\chi^2 = 0.29$, d.f. = 1, $p = 0.59$). Indeed, for the dropped item experiment we observed the opposite pattern: people were more likely to offer help when they were alone (94/220, 42.7%, helped) compared to when they were walking with another person (36/178, 20.2%, helped; $\chi^2$-test, $\chi^2 = 21.6$, d.f. = 1, $p < 0.001$). Overall, there was no effect of urbanicity on the likelihood of receiving help either when people were with a friend (City: 39/161, 24.2%, helped; Town: 13/63, 20.6%, helped; $\chi^2$-test, $\chi^2 = 0.16$, d.f. = 1, $p =$

0.69), or when they were alone (City: 69/173, 39.9%, helped; Town: 37/91, 40.7%, helped; $\chi^2$-test, $\chi^2 < 0.001$, d.f. = 1, $p = 1$).

## 4. Discussion

Our data do not support the idea that urbanicity is associated with reduced generalized prosociality. Instead, most of the variation in whether help was offered was explained by neighbourhood wealth, with help being more forthcoming in higher-wealth neighbourhoods. Our findings contrast with previous theories and empirical studies on the role of urbanicity in shaping prosocial behaviour (see [21,50] for reviews) but support more recent work that has implicated relative deprivation as

being key to understanding variation in generalized prosociality and collective behaviour [34–36,51–54]. The effect of neighbourhood wealth on the tendency to help a stranger might help to explain why previous studies have reported decreased prosociality among urban dwellers. Although the pattern in the UK is quite mixed, there is quantitative evidence that rural areas are typically less deprived than more urban locations (a pattern than we also observe in the locations selected for this study) [42]. Previous findings that generalized cooperation is reduced in urban areas might, therefore, be masking the underlying instrumental variable, which relates to deprivation rather than to population density.

These data from a battery of real-world experiments lend more weight to the hypothesis that relative deprivation is negatively associated with generalized trust and prosociality. Several other studies have reported a similar pattern [34,52–61]. In experimental settings, exposure to harsh environments has been associated with an increased willingness to defect in a Prisoner's Dilemma game, and a reduced tendency to send money to a partner in a Dictator Game [36,62,63], and experimentally induced financial deprivation can increase the willingness to cheat for financial rewards [64]. Studies using the lost letter technique to measure prosociality have consistently reported increased return rates from higher-wealth neighbourhoods [34,35,56]. Large-scale, survey-based studies report similar findings. One recent study using more than 30 000 observations based on nationally representative samples concluded that high socio-economic status was associated with increased willingness to donate to charity, to volunteer to help, to contribute a higher proportion of income to charity, and to choose the prosocial option in an economic game [59] (see also [58]), while another large study (a total sample of greater than 60 000 and with participants from more than 30 countries) reported positive effects of household income on tendency to volunteer or to donate to charity ([60], but see [65], who performed a similar study, obtaining a null result). Finally, a study using data from more than 40 000 responses to the World Values Survey and the European Values Study, respectively, also suggests a negative link between exposure to environmental harshness and the tendency to invest in cooperative behaviour [52].

Nevertheless, other studies have reported negative effects of socio-economic status on prosocial and ethical behaviour [66–72]. We cannot account for the apparently contradictory findings in this field though we note that some of these earlier studies have been based on relatively small undergraduate samples [67,69] and several key results have failed to replicate in large-N, pre-registered replication attempts [73]. Other work has not found a main effect of social class on prosocial behaviour but has shown, instead, that the effect is moderated by third variables. For example, one study found that the effect of social class on social behaviour depends on whether the behaviour takes place in private or public [74], though we note that the effects of the interaction between class and anonymity on giving was inconsistent across different experiments in the study mentioned. Another study argued that the effect of social class on prosocial behaviour depends on whether lower-status individuals are exposed to high economic inequality [75], though two subsequent studies have failed to replicate this effect using similarly large datasets [58,60].

We note that our study differs from much of the work reported above in that we measured behaviour in the real world, rather than via online surveys or in the context of experimental games. In this true field experiment, it is possible that participants were also affected by the environment in which the behaviour was measured. For example, people are more likely to violate prosocial norms of collective behaviour where there is evidence that others do the same [76]—evidence that these norms are violated (e.g. litter and other indices of disorder) is typically higher in more deprived communities [77]. Other work has shown that prosocial behaviour is more likely when people have access to green space [78]. Importantly, access to green space is often lower in more deprived areas [77,79] or else is used less frequently by residents due to perceptions of being inaccessible or unsafe [79,80]. With our current dataset, it is impossible for us to determine whether the link between helping behaviour and deprivation pertains to the deprivation experienced by the participants or, alternatively, to the nature of the environment in which the studies were conducted, though we note that these explanations are not mutually exclusive.

Assuming that the negative association between neighbourhood levels of deprivation and the reduced tendency to help a stranger does exist, it begs the question as to how such a relationship might arise. There are several plausible routes by which deprivation might lead to reduced tendency to help a stranger. One of the most plausible routes might be through the effects that environmental harshness or unpredictability has on the tendency to invest to achieve larger rewards in the future, rather than taking immediately available, smaller pay-offs now. Investing to help a stranger has this incentive structure, where any downstream benefits of the helpful action are typically delayed and/or uncertain [38,81,82]. The effect of reduced income or lowered neighbourhood quality on the tendency to help a stranger could also be mediated by reduced social capital and generalized trust [83–86]. For example, a natural experiment in Russia found that a 10% decrease in average national income following the 2009 recession was associated with a 5% decrease in social trust [87]. Given that one of the help measures in our study (dropped item) involved a live interaction with a stranger, trust may well have been a relevant concern for people deciding whether to help or not. Another potential explanation for the link between adversity and willingness to help a stranger might be the role of material security and how this impacts the scale at which people cooperate [88–91]. In brief, this hypothesis predicts that as material security increases, people are more able to expand their social network, offering impartial help and cooperation to people beyond their core social group of known and regular interaction partners. We note that a prediction that derives from all these hypotheses (albeit one that we cannot test with our data) is that higher indices of deprivation in the UK will only affect the willingness to help a stranger, but not the willingness to help others that are part of one's existing community. Some empirical work supports the hypothesis that exposure to adversity or low socio-economic status is associated with an increased tendency to help friends or in-group members [70,71] though this hypothesis deserves further empirical attention.

We found mixed support for the hypothesis that help would be more likely when participants were directly requested to help. In the dropped item experiment, the direct request increased the likelihood of receiving help, whereas in the lost letter experiment, there was no difference in return rates across the direct request and no request conditions. These patterns may stem from the fact that the direct request in the

dropped item experiment was made face-to-face, whereas in the lost letter experiment, the request was made remotely. The perceived costs of refusing to help when asked are likely to be higher in a face-to-face interaction, where the helper can be seen and potentially identified by the requester [26], and other work has shown that people are more cooperative in face-to-face interactions and/or when their name and picture are shown to others [26,30]. The difference we observe between the lost letter and the dropped item experiment helps to further quantify when and how direct requests might elicit help. Specifically, our study suggests that direct requests might increase helpful behaviour not because this reduces the diffusion of responsibility, but because a direct request increases the perceived reputation costs of refusing to help. This hypothesis could be explored in further experimental work.

We expected individuals in a group to help more than lone individuals as the presence of others would create the opportunity to accrue reputation benefits (e.g. [33,92,93]). However, in contrast with our expectations, we recorded higher helping from lone individuals than from individuals in groups in one of our experiments. One possibility is that individuals in social groups might be under greater perceptual load (e.g. in conversation with their acquaintance) which means that they do not note the helping opportunity. Another possibility is that people in groups experience greater perceived costs from pursuing an independent course of action (stopping to help an experimenter) that requires them to temporarily deviate from the group action, and also to impose the time costs

associated with helping onto their acquaintance. We do not know of any study that addresses these latter possibilities empirically.

These results contribute to our understanding of the factors affecting variation in human cooperation. These data challenge the folk view that city dwellers are less cooperative than town dwellers and show that this variation may be a by-product of the association between urbanicity and deprivation. More generally, this study supports the hypothesis that deprivation reduces the willingness to extend impartial norms of cooperation towards strangers.

Ethics. No personal identifying information was collected about any of the subjects in these experiments. We could not obtain informed consent from participants due to the nature of the data collection. Experiments fell under the ethical remit of project 3720/001 (UCL Ethics Board). Metadata associated with neighbourhoods are publicly available on UK government websites.

Data accessibility. Data and code to reproduce the main analyses are available at the following URL: https://osf.io/cmdfk/.

Authors' contributions. N.R. conceived the study. N.R. and E.Z. designed experiments and E.Z. collected data. N.R. analysed the data and both authors wrote the paper.

Competing interests. We declare we have no competing interests.

Funding. This study was funded by a BA-Leverhulme Research Grant to N.R.

Acknowledgements. We would like to thank Richard McElreath for his help with Bayesian analyses, Henrik Singmann for other statistical advice, and Dan Nettle for helpful comments on the thesis chapter which predated this manuscript.

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
