## [Reviewer comments · Proceedings of the Royal Society B: Biological Sciences]

Review History

RSPB-2020-1359.R0 (Original submission)

Review form: Reviewer 1 (Kimmo Eriksson)

Recommendation

Accept with minor revision (please list in comments)

Scientific importance: Is the manuscript an original and important contribution to its field?
Excellent

General interest: Is the paper of sufficient general interest?
Excellent

Quality of the paper: Is the overall quality of the paper suitable?
Good

Is the length of the paper justified?
Yes

Should the paper be seen by a specialist statistical reviewer?
Yes

Do you have any concerns about statistical analyses in this paper? If so, please specify them explicitly in your report.

Yes

It is a condition of publication that authors make their supporting data, code and materials available - either as supplementary material or hosted in an external repository. Please rate, if applicable, the supporting data on the following criteria.

Is it accessible?

No

Is it clear?

N/A

Is it adequate?

N/A

Do you have any ethical concerns with this paper?

No

Comments to the Author

This paper reports field experiments on prosocial behavior: posting letters, helping with dropped items, and stopping to let someone cross the road. The experiments were carried out across locations in the UK. The locations were selected to vary across two dimensions: urbanicity (cities vs. towns) and affluence (high wealth vs. low wealth). The authors examined how these dimensions influenced the rate of prosocial behavior and found a significant and substantial positive effect of affluence but no effect of urbanicity. Overall this is a good paper with very interesting results. I have only a few issues with it that I hope should be easily fixed.

1. It is surprising that the title does not include the positive finding but only the negative finding.
2. Similarly in the abstract it is not clear that your study was designed explicitly to test both the urbanicity and affluence hypotheses.
3. After reading the abstract I did not understand that these were proper field experiments and not survey experiments (which may also be considered "real-world" as opposed to econ lab games). I don't think you measured "willingness", you measured actual behavior. Do not undersell this strength of your study.
4. The crucial weakness of the current manuscript is the poor description of the statistical analysis. I think Fig 2 is lovely and compellingly illustrates the results. But the meaning of the values in Fig 1 is beyond me and I wonder if even the authors know what it is; at least, I found no attempt to explain what the unit of the values is. This goes for the text too: what does it mean that the effect of affluence was estimated to 0.82? Moreover, what does an "89 % interval" mean? Only much later in the paper do you mention that you use a Bayesian approach, but this must be explained much earlier and in sufficient detail (including prior distributions) for readers to decode your results. And you need to write out the full logistic regression equation so that readers can understand what you are in fact estimating.
5. To be publishable, I think the discussion section must heads-on address and attempt to reconcile your findings of greater prosociality in affluence neighborhoods with the body of research, reviewed by Piff and Robinson (which you cite), finding more prosocial behavior in the lower social class. I guess most of their research is US based; could it be that your finding is specific to the UK and would not replicate in the US? I also guess most of their research is on giving experiments and not on real-life behavior in different neighborhoods; could it be that the

crucial parameter in your experiment is not who but where, that is, that the same individuals behave differently depending of whether they are in a rich or deprived neighborhood?

Review form: Reviewer 2

Recommendation

Accept with minor revision (please list in comments)

Scientific importance: Is the manuscript an original and important contribution to its field?

Excellent

General interest: Is the paper of sufficient general interest?

Excellent

Quality of the paper: Is the overall quality of the paper suitable?

Excellent

Is the length of the paper justified?

Yes

Should the paper be seen by a specialist statistical reviewer?

Yes

Do you have any concerns about statistical analyses in this paper? If so, please specify them explicitly in your report.

No

It is a condition of publication that authors make their supporting data, code and materials available - either as supplementary material or hosted in an external repository. Please rate, if applicable, the supporting data on the following criteria.

Is it accessible?

Yes

Is it clear?

Yes

Is it adequate?

Yes

Do you have any ethical concerns with this paper?

No

Comments to the Author

The authors set out to examine an interesting question, exploring whether prosocial tendencies of humans are better explained by whether they live in a town or a city, or by the deprivation level of the neighborhood they live in. Additionally, the authors look whether a direct or indirect request to help increases prosocial behaviors, and whether being in a group or being alone in- or decreased helping behavior.

Overall, I think this is a really interesting study, which seems to be of high scientific value, and is nicely written. Thus, I recommend to accept this paper for publication with only asking for a few

minor revisions:

- In the introduction, the authors give a good overview of the literature investigating the factors city/town and deprivation, however, the literature overview for the factors population density, indirect/direct request, and alone/group is rather spare. I would suggest the authors to give some background for all predictors they are investigating.
- I think the analysis reported deserves some more details. It is not clear from the main manuscript which factors exactly were entered in the model. Where there any interactions that were of interest? Can you explain the steps of the analysis in more detail?

Decision letter (RSPB-2020-1359.R0)

04-Sep-2020

Dear Professor Raihani:

Your manuscript has now been peer reviewed and the reviews have been assessed by an Associate Editor. The reviewers' comments (not including confidential comments to the Editor) and the comments from the Associate Editor are included at the end of this email for your reference. As you will see, the reviewers and the Editors have raised some concerns with your manuscript and we would like to invite you to revise your manuscript to address them.

Research ethics:

Use of animals and field studies:

It is a condition of publication that you make available the data and research materials supporting the results in the article. Please see our Data Sharing Policies (<https://royalsociety.org/journals/authors/author-guidelines/#data>). Datasets should be deposited in an appropriate publicly available repository and details of the associated accession number, link or DOI to the datasets must be included in the Data Accessibility section of the article (<https://royalsociety.org/journals/ethics-policies/data-sharing-mining/>). Reference(s) to datasets should also be included in the reference list of the article with DOIs (where available).

Please submit a copy of your revised paper within three weeks. If we do not hear from you within this time your manuscript will be rejected. If you are unable to meet this deadline please let us know as soon as possible, as we may be able to grant a short extension.

Best wishes,
Dr John Hutchinson, Editor
<mailto:proceedingsb@royalsociety.org>

Associate Editor

Board Member: 1

Comments to Author:

This paper reports the results of real-world experiments measuring willingness to cooperate. Real world experiments are rare and hard to conduct. It is extremely interesting, well executed and well explained and in my view will be of interest to a broad audience. I agree with the reviewers, however, that the statistical analysis needs more explanation, and a title change might emphasise the novelty of the results.

Reviewer(s)' Comments to Author:

Referee: 1

Comments to the Author(s)

This paper reports field experiments on prosocial behavior: posting letters, helping with dropped items, and stopping to let someone cross the road. The experiments were carried out across locations in the UK. The locations were selected to vary across two dimensions: urbanicity (cities vs. towns) and affluence (high wealth vs. low wealth). The authors examined how these dimensions influenced the rate of prosocial behavior and found a significant and substantial positive effect of affluence but no effect of urbanicity. Overall this is a good paper with very interesting results. I have only a few issues with it that I hope should be easily fixed.

1. It is surprising that the title does not include the positive finding but only the negative finding.
2. Similarly in the abstract it is not clear that your study was designed explicitly to test both the urbanicity and affluence hypotheses.
3. After reading the abstract I did not understand that these were proper field experiments and not survey experiments (which may also be considered "real-world" as opposed to econ lab games). I don't think you measured "willingness", you measured actual behavior. Do not undersell this strength of your study.
4. The crucial weakness of the current manuscript is the poor description of the statistical analysis. I think Fig 2 is lovely and compellingly illustrates the results. But the meaning of the values in Fig 1 is beyond me and I wonder if even the authors know what it is; at least, I found no attempt to explain what the unit of the values is. This goes for the text too: what does it mean that the effect of affluence was estimated to 0.82? Moreover, what does an "89 % interval" mean? Only much later in the paper do you mention that you use a Bayesian approach, but this must be explained much earlier and in sufficient detail (including prior distributions) for readers to decode your results. And you need to write out the full logistic regression equation so that readers can understand what you are in fact estimating.
5. To be publishable, I think the discussion section must heads-on address and attempt to reconcile your findings of greater prosociality in affluence neighborhoods with the body of research, reviewed by Piff and Robinson (which you cite), finding more prosocial behavior in the lower social class. I guess most of their research is US based; could it be that your finding is specific to the UK and would not replicate in the US? I also guess most of their research is on giving experiments and not on real-life behavior in different neighborhoods; could it be that the crucial parameter in your experiment is not who but where, that is, that the same individuals behave differently depending of whether they are in a rich or deprived neighborhood?

Referee: 2

Comments to the Author(s)

The authors set out to examine an interesting question, exploring whether prosocial tendencies of humans are better explained by whether they live in a town or a city, or by the deprivation level

of the neighborhood they live in. Additionally, the authors look whether a direct or indirect request to help increases prosocial behaviors, and whether being in a group or being alone in- or decreased helping behavior.

Overall, I think this is a really interesting study, which seems to be of high scientific value, and is nicely written. Thus, I recommend to accept this paper for publication with only asking for a few minor revisions:

- In the introduction, the authors give a good overview of the literature investigating the factors city/town and deprivation, however, the literature overview for the factors population density, indirect/direct request, and alone/group is rather sparse. I would suggest the authors to give some background for all predictors they are investigating.

- I think the analysis reported deserves some more details. It is not clear from the main manuscript which factors exactly were entered in the model. Where there any interactions that were of interest? Can you explain the steps of the analysis in more detail?

Author's Response to Decision Letter for (RSPB-2020-1359.R0)

See Appendix A.

Decision letter (RSPB-2020-1359.R1)

14-Sep-2020

Dear Professor Raihani

I am pleased to inform you that your manuscript entitled "Neighbourhood wealth, not urbanicity, predicts prosociality towards strangers" has been accepted for publication in Proceedings B. Congratulations!!

Open Access

Paper charges

Sincerely,

Dr John Hutchinson

Associate Editor:

Comments to Author:

The authors have done an excellent job of responding to the referees' comments and the paper reads better, particularly the stats which required some extra detail. I like the title change and think this better reflects the findings. On the whole I really like this paper and am confident that the paper will reach a broad audience.

Appendix A

Responses to Reviewers

We would like to thank the Board Member and both editors for the extremely constructive feedback, which we have incorporated into the revised manuscript. This has been one of the most pleasant and genuinely helpful peer-review experiences that either of us has experienced. Thank you for that! We note all the point by point responses to the reviewer comments below, in blue. Excerpts from the revised manuscript are shown in red.

Associate Editor

Board Member: 1

Comments to Author:

This paper reports the results of real-world experiments measuring willingness to cooperate. Real world experiments are rare and hard to conduct. It is extremely interesting, well executed and well explained and in my view will be of interest to a broad audience. I agree with the reviewers, however, that the statistical analysis needs more explanation, and a title change might emphasise the novelty of the results.

Thanks for this – we have changed the title to:

“Neighbourhood wealth, not urbanicity, predicts prosociality towards strangers”

We have also added more explanation of the statistical methods (please see below).

Reviewer(s)' Comments to Author:

Referee: 1

Comments to the Author(s)

This paper reports field experiments on prosocial behavior: posting letters, helping with dropped items, and stopping to let someone cross the road. The experiments were carried out across locations in the UK. The locations were selected to vary across two dimensions: urbanicity (cities vs. towns) and affluence (high wealth vs. low wealth). The authors examined how these dimensions influenced the rate of prosocial behavior and found a significant and substantial positive effect of affluence but no effect of urbanicity. Overall this is a good paper with very interesting results. I have only a few issues with it that I hope should be easily fixed.

Thank you so much for the positive feedback.

1. It is surprising that the title does not include the positive finding but only the negative finding.

We have now changed the title as noted above.

2. Similarly in the abstract it is not clear that your study was designed explicitly to test both the urbanicity and affluence hypotheses.

We have added a line in the abstract as follows:

However, it is not known whether a general tendency towards prosocial behaviour varies across the urban-rural gradient, or whether other factors such as neighbourhood wealth might be more predictive of variation in prosocial behaviour.

3. After reading the abstract I did not understand that these were proper field experiments and not survey experiments (which may also be considered “real-world” as opposed to econ lab games). I don’t think you measured “willingness”, you measured actual behavior. Do not undersell this strength of your study.

Thanks for this – we re-worded the abstract slightly in line with your suggestion.

4. The crucial weakness of the current manuscript is the poor description of the statistical analysis. I think Fig 2 is lovely and compellingly illustrates the results. But the meaning of the values in Fig 1 is beyond me and I wonder if even the authors know what it is; at least, I found no attempt to explain what the unit of the values is. This goes for the text too: what does it mean that the effect of affluence was estimated to 0.82? Moreover, what does an “89 % interval” mean? Only much later in the paper do you mention that you use a Bayesian approach, but this must be explained much earlier and in sufficient detail (including prior distributions) for readers to decode your results. And you need to write out the full logistic regression equation so that readers can understand what you are in fact estimating.

Sorry about this – and you were completely right to point this out. A small part of the confusion stems from us having submitted a manuscript with the Intro, Results, Discussion, Methods format. We have now altered the manuscript so that Methods follow Intro as this is more conventional and easier to understand. In doing this, we removed some redundant information from the introduction that was repeated in the Methods.

However, you were also right that we did not present or explain the output of our models well. We originally presented the untransformed mean estimates from our model, which added to the confusion. We have now changed Figure 1 so that it shows the back-transformed means (which can be understood as binomial probabilities of receiving help). We have added more information in the statistical methods section (lines 225 – 244) to explain how the models were run, and the priors we chose. We have changed the results section so that we report the back-transformed posterior mean probabilities (lines 253-261). We also include far more text in the legend to Figure 1 (line 265-277) to explain how a reader should interpret the figure. We are also open to removing the figure if the reviewer and editor think that would be simpler, though we think it is helpful / interesting to show some of the regional variation in the probability of receiving help, which this figure illustrates. We would also like to note that all data and code to reproduce the analyses are available at <https://osf.io/cmdfk/>.

5. To be publishable, I think the discussion section must heads-on address and attempt to reconcile your findings of greater prosociality in affluence neighborhoods with the body of research, reviewed by Piff and Robinson (which you cite), finding more prosocial behavior in the lower social class. I guess most of their research is US based; could it be that your finding is specific to the UK and would not replicate in the US? I also guess most of their research is on giving experiments and not on real-life behavior in different neighborhoods; could it be that the crucial parameter in your experiment is not who but where, that is, that the same individuals behave differently depending of whether they are in a rich or deprived neighborhood?

Thanks for this suggestion. We have now added two more paragraphs in the discussion (lines 349 – 378) explicitly addressing this point. We agree that where the studies are conducted could affect helping behaviour in our tasks and we raise this possibility along with supporting literature. We also discuss in more detail the mixed pattern of results in this field more generally. We have also updated our literature review in this section to include some recently-published work.

Referee: 2

Comments to the Author(s)

The authors set out to examine an interesting question, exploring whether prosocial tendencies of humans are better explained by whether they live in a town or a city, or by the deprivation level of the neighborhood they live in. Additionally, the authors look whether a direct or indirect request to help increases prosocial behaviors, and whether being in a group or being alone in- or decreased helping behavior.

Overall, I think this is a really interesting study, which seems to be of high scientific value, and is nicely written. Thus, I recommend to accept this paper for publication with only asking for a few minor revisions:

Thank you so much for the positive feedback.

- In the introduction, the authors give a good overview of the literature investigating the factors city/town and deprivation, however, the literature overview for the factors population density, indirect/direct request, and alone/group is rather spare. I would suggest the authors to give some background for all predictors they are investigating.

Thank you for pointing this out. We have now included some additional text in the introduction (lines 65-77), methods (lines 138-141) to explain the justification for these factors, including supporting references. In addition, we report some new analyses to further explore the effects of direct request and audience presence on helping behaviour (lines 291-295 and 304-307). Furthermore, we also added some text to the discussion reporting on the effects of direct request and audience presence (lines 407-420).

- I think the analysis reported deserves some more details. It is not clear from the main manuscript which factors exactly were entered in the model. Where there any interactions that were of interest? Can you explain the steps of the analysis in more detail?

We completely agree with this critique. Please see the response to Reviewer 1 above where we describe the revisions we made. We would also like to note that all data and code to reproduce the analyses are available at <https://osf.io/cmdfk/>.